# The Clinical Significance of Circulating Tumor DNA for Minimal Residual Disease Identification in Early-Stage Non-Small Cell Lung Cancer

**DOI:** 10.3390/life13091915

**Published:** 2023-09-15

**Authors:** Alberto Verlicchi, Matteo Canale, Elisa Chiadini, Paola Cravero, Milena Urbini, Kalliopi Andrikou, Luigi Pasini, Michele Flospergher, Marco Angelo Burgio, Lucio Crinò, Paola Ulivi, Angelo Delmonte

**Affiliations:** 1Medical Oncology Department, IRCCS Istituto Romagnolo per lo Studio dei Tumori (IRST) “Dino Amadori”, 47014 Meldola, Italy; alberto.verlicchi@irst.emr.it (A.V.); paola.cravero@irst.emr.it (P.C.); kalliopi.andrikou@irst.emr.it (K.A.); michele.flospergher@irst.emr.it (M.F.); marco.burgio@irst.emr.it (M.A.B.); lucio.crino@irst.emr.it (L.C.); angelo.delmonte@irst.emr.it (A.D.); 2Biosciences Laboratory, IRCCS Istituto Romagnolo per lo Studio dei Tumori (IRST) “Dino Amadori”, 47014 Meldola, Italy; elisa.chiadini@irst.emr.it (E.C.); milena.urbini@irst.emr.it (M.U.); luigi.pasini@irst.emr.it (L.P.); paola.ulivi@irst.emr.it (P.U.)

**Keywords:** non-small cell lung cancer, minimal residual disease, circulating tumor DNA

## Abstract

Lung cancer (LC) is the deadliest malignancy worldwide. In an operable stage I–III patient setting, the detection of minimal residual disease (MRD) after curative treatment could identify patients at higher risk of relapse. In this context, the study of circulating tumor DNA (ctDNA) is emerging as a useful tool to identify patients who could benefit from an adjuvant treatment, and patients who could avoid adverse events related to a more aggressive clinical management. On the other hand, ctDNA profiling presents technical, biological and standardization challenges before entering clinical practice as a decisional tool. In this paper, we review the latest advances regarding the role of ctDNA in identifying MRD and in predicting patients’ prognosis, with a particular focus on clinical trials investigating the potential of ctDNA, the technical challenges to address and the biological parameters that influence the MRD detection.

## 1. Introduction

Lung cancer (LC) is the second most commonly diagnosed cancer and the leading cause of cancer-related death, representing approximately 1 in 10 cancers diagnosed and 1 in 5 cancer-related deaths [1]. Non-small cell lung cancer (NSCLC) represents 85% of all lung cancers, including adenocarcinoma, squamous cell carcinoma and large-cell carcinoma [2]. Patient overall survival (OS) is mainly dependent on the disease stage at the time of diagnosis, with a 57.5% median 5-year OS rate in patients with resectable stages I–IIIa (American Joint Committee on Cancer (AJCC) 7th edition). A more advanced stage at diagnosis and tumor recurrence after surgery (occurring in 30–70% of patients) negatively influence 5-year OS, ranging from 49% for stage Ib to 20% for stage IIIa in patients with recurrent malignancy [3]. The presence of free nucleic acids in the blood plasma was first reported in 1948 [4]. Cell-free DNA (cfDNA) is released in the peripheral blood circulation during cellular processes such as apoptosis and necrosis or by active secretion [5]. Fragment sizes vary from 40 to 200 base pairs (bp), with a peak at 166 bp, suggesting they are wrapped in nucleosomes cut by endonucleases, and 10 bp fragments, corresponding to a turn of DNA helix around the core histone [6,7,8]. cfDNA can also be isolated in other body fluids, such as saliva, urine and pleural and cerebrospinal fluids, with half-life ranges between 16 min and 2.5 h, for which the main clearance mechanisms occur by active nuclease activity or excretion by urine [9]. Circulating tumor DNA (ctDNA) is a fraction of cfDNA, which is commonly indicated as the fraction of tumor DNA released in the body fluids, ranging from 0.05% to 90% in the background of the whole cfDNA [10,11,12]. ctDNA concentration is more elevated in cancer patients than in healthy subjects, as well as in specific tumor histologies (i.e., bladder cancer, colorectal cancer and ovarian cancer), and its fraction in the cfDNA is generally influenced by factors such as tumor volume and vascularization, and by anticancer treatments [13,14,15,16]. Moreover, Bettegowda and colleagues demonstrated that ctDNA concentration also depends on staging, as detectable ctDNA was 47%, 55%, 69% and 82% in patients with different tumors at stage I, II, III and IV, respectively [17]. cfDNA clearance is performed in organs, such as kidney, spleen, lymph nodes and liver, or in the blood by circulating enzymes, such as DNase I, plasma factor VII-activating protease (FSAP) and factor H. [18,19]. In particular, up to 84.7% of circulating nucleosomes are cleared by liver Kupffer cells and spleen macrophages [20]. Considering its short half-life and the continuous active clearance mechanisms, ctDNA is considered a “real-time” snapshot of tumor molecular heterogeneity and tumor burden. To date, ctDNA has entered clinical practice in treatment guidance for advanced NSCLC oncogene addiction, therapeutic benefit and disease monitoring and in survival prediction [21,22,23,24]. More recently, ctDNA longitudinal monitoring and molecular profiling brought promising results as a useful clinical tool for predicting minimal residual disease (MRD) in early-stage and locally advanced NSCLC patients treated with curative intent [25,26]. In this review, we summarize the most recent advances in the clinical utility of ctDNA in the early stage NSCLC setting, with particular focus on clinical trials evaluating the role of ctDNA, and then we explore technical and biological factors that limit ctDNA as a clinical practice biomarker.

## 2. Detection of MRD through Liquid Biopsy

The most challenging issue in detecting MRD remains the biomarker selection, and more importantly a consequent sensitivity rate able to identify and discriminate patients with a higher risk of relapse. Most recent advances have focused on ctDNA molecular profiling, highlighting it as the most promising biomarker to detect MRD, and the improvements in molecular and computational biology have reached good results in enhancing sensitivity and specificity, reducing sequencing errors when increasing limit of detection (LOD) [27,28]. From a clinical laboratory point of view, the LOD is ideally the ctDNA concentration at which 95% of clinical samples are positive [29]. Genomic analysis of cfDNA also includes the calculation of the circulating tumor fraction DNA. Mutant allele fraction (MAF) of a somatic alteration has been considered as the absolute estimation of tumor fraction, but MAF and ctDNA also have been considered in relation to define the tumor fraction [30,31,32,33]. To date, the detection and molecular profiling of ctDNA is performed through polymerase chain reaction (PCR) or next-generation sequencing (NGS), for the identification of tumor-specific mutations, copy number or structural alterations, or epigenetic patterns of solid tumors [34,35,36,37].

### 2.1. Digital Polymerase Chain Reaction

Digital polymerase chain reaction (dPCR) is the most sensitive among the PCR-based application methodologies (i.e., real-time quantitative PCR: qPCR; amplification refractory mutation system-PCR: ARMS-qPCR; nested-qPCR) [38]. It is based on the dilution of the template to a certain concentration and dispersion in micro-reaction units before amplification, with theoretical one DNA target sequence in each unit [39]. The most frequently used methodology is droplet-dPCR (ddPCR), in which the amplification unit is represented by a single droplet of diluted template. This technique has achieved an LOD of 0.001% in detecting DNA mutations in tissue, while it is 0.1% in plasma samples [40,41,42,43]. While ddPCR is very useful to detect hotspot mutations of high prevalence genes in cancer, it is very difficult to multiplex different probes to build a ddPCR-based panel for MRD detection; it has relatively high costs and it needs highly-specific probes to reduce cross-reactivity [44]. Moreover, designing a panel to detect MRD in liquid biopsy of NSCLC patients could also be time-consuming, especially if a “tumor-informed” approach is adopted (see Section 2.2), as ddPCR has the limitation that mutations must be known in advance [45]. More importantly, multiplexing of different probes is challenging, as sensitivity of the methodology is mainly based on the different binding affinities of mutant and wild-type alleles [46]. Consequently, this methodology is more suitable for the detection of activating known mutations in NSCLC (e.g., EGFR mutations) or hotspot resistance mutations to tyrosine kinase inhibitors (e.g., *ALK* C1156Y, I1171N or *EGFR* C797X) [45,46,47]. Xu and colleagues built a 37-gene PCR-based approach based on forward primers and blocker probes to amplify the mutant allele in a selective manner, namely PEAC. With this methodology, the authors identified a tumor relapse in a stage Ib NSCLC patient 7 months after surgery, two months before clinical and radiological relapse [48]. Interestingly, a recent paper used a multiplexed dPCR to develop a methylation-specific PCR platform; the results show that a four-biomarker panel was detectable in 72 patients with CT scan indeterminate nodules, with 90% sensitivity and 82% specificity [49]. These data suggest that a dPCR-selected biomarker panel could be useful to detect NSCLC early, and the clinical sensitivity in detecting MRD could be tested.

### 2.2. Next-Generation Sequencing

Massive parallel sequencing has been incorporated in many laboratory routine approaches for cancer molecular profiling. Several NGS methodologies have been tested in liquid biopsy for early cancer detection, molecular profiling, prognosis and therapy monitoring for solid malignancies [34,50,51,52]. Most ctDNA profiling approaches are based on “tumor-informed” analysis, which has the aim of detecting and monitoring genomic alterations firstly identified in tumor tissue. Even though the tracking of specific mutations previously identified would require less material and could take advantage of a more sensitive methodology, it could be less informative than “tumor non-informed” approaches [29]. In fact, tumor non-informed approaches are based on the sequencing of the sole liquid biopsy, without prior knowledge of molecular profile of the tissue. While tumor-informed approaches have the highest analytical sensitivity, a tumor non-informed approach could be used for wide genotyping and early detection of emerging mutations, with lower sensitivity for the multiple testing.

Generally, NGS methodology is considered to reach a lower sensitivity than dPCR, even though ultradeep sequencing was revealed to reach 75% sensitivity for blinded-to-tissue plasma genotyping and 100% specificity in tissue wild-type tumors, and to detect the same mutations results on EGFR and KRAS mutations with respect to dPCR in liquid biopsies in 21 out of 22 patients [53].

#### 2.2.1. Targeted Sequencing

Targeted sequencing is based on the amplification and sequencing of a panel of selected genes o genomic regions. One targeted approach tested the potential of molecular amplification pools (MAPs) in reducing NGS sequencing errors, with a sensitivity of 98.5% and a specificity of 98.9%, with similar accuracy with respect to ddPCR. MAPs is a methodology used to reduce sequencing errors by dividing cfDNA in two pools, performing a targeted amplification and by statistically comparing the sequencing results with large numbered molecular pools to eliminate stochastic and recurrent errors [27].

Amplicon-based or hybrid capture NGS are commonly used for sequencing targeted regions. These approaches are mainly related to NGS gene panels analyzing a number of targeted cancer-related loci (kb), but it is possible to sequence the entire exome (50 Mb) [36]. Prior to sequencing, amplicon-based assays target hundreds of PCR amplicons, while hybrid capture increase the genomic region of interest [54,55,56]. Considering this possibility, this methodology could be used in a patient-specific approach for cancer profiling after the sequencing of tumor tissue, e.g., cancer-personalized profiling using deep sequencing (CAPP-seq) [57,58,59]. In this setting, Chabon and colleagues developed a machine learning method termed “lung cancer likelihood in plasma” (Lung-CLiP), able to discriminate early-stage lung cancer patients form risk-matched controls [60]. Moreover, a recent approach enhanced the limit of detection of sequencing in the ppm range in patients with large B cell lymphoma, also demonstrating the feasibility of the method in solid malignancies; in particular, this method, namely PhasED-Seq, identified 25% of patients with undetectable ctDNA using CAPP-seq, who had a worse outcome. The approach is based on the enrichment of multiple somatic mutations found in ***cis*** (on the same strand of DNA) in individual DNA fragments, to reduce the sequencing errors, as two molecular alterations provide two independent nonreference events. By focusing on mutations occurring at a distance of <170 nt (the length of a core nucleosome and DNA linker in ctDNA) and by leveraging whole-genome sequencing (WGS) data from 2538 tumors, the authors highlighted cancer-specific enriched regions and were able to identify mutations at high sensitivity with low error rates [61]. Another capture panel-based approach is targeted error correction sequencing (TEC-Seq), tested for cancer diagnosis by analyzing 81 kb of the most 58 highly mutated genes across different cancers, and that made it possible to identify patients with lung cancer at stage I or II with a sensitivity of 59%. Using a COSMIC database somatic mutation search, a set of 55 genes to have at least an alteration in three of four cancer patients was identified, to which 3 commonly altered genes in hematological disorders. The methodology focused on redundant deep sequencing (~30,000×) of targeted regions of the selected genes (80,930 bases). After sequencing error corrections, the authors reached an analytical sensitivity of 97.4%, with a specificity of >99.99% [62]. The molecular profiling of ctDNA was used to track the phylogenetic evolution of NSCLC in the TRACERx study, in which a multi-region exome sequencing was performed. Single nucleotide variants (SNV) above 0.1% were detected with <99% sensitivity in an early stage cohort of NSCLC patients. In a subgroup of 24 patients followed by longitudinal ctDNA analyses, a patient-specific multiplex PCR assay panel was able to anticipate relapse confirmed by CT scan with a median of 70 days (range 10–346 days) [63]. In this context, a recent systematic review by Verzè and colleagues focused on 13 studies detecting MRD by ctDNA analysis concluded that this tool is able to precede radiographic/clinical recurrence by a mean of 5.5 months. Moreover, this study highlighted a concordance rate ranging from 43.7% to 92.2% between ctDNA and tumor tissue using different molecular biology approaches, with a range of detection MRD ranging from 6 to 46% [64]. Moreover, one study part of this systematic review enrolling 21 patients and analyzing pre-, peri- and postoperative plasma samples showed that positive ctDNA in an early postoperative temporal window is associated with shorter PFS (*p* = 0.013) and OS (*p* = 0.004) [65].

Targeted sequencing methodologies are the most sensitive among NGS ones, especially considering the great efforts in terms of error rates reduction and computational biology improvements. On the other hand, targeted panels have the limitation of analyzing a set of genes, with the possibility to not detect a genomic alteration of the tumor. “Tumor-informed” approaches could address this matter, but tumor heterogeneity, prolonging of turnaround times and augmenting of costs remain major limitations.

#### 2.2.2. Whole-Genome and Whole-Exome Sequencing

WGS and whole-exome sequencing (WES) potentially discover a much higher number of mutations than targeted sequencing. On the other hand, these approaches carry a large number of background sequencing errors, and this represents a major limitation especially in the MRD setting, in which the expected ctDNA quantity is low. Moreover, the high costs and turnaround time make them difficult to be feasible for routine cancer molecular profiling [29].

A schematic representation of the utility of ctDNA for MRD detection before radiologic progression is provided in Figure 1.

## 3. Minimal Residual Disease in NSCLC Adjuvant Setting

For early-stage NSCLC, defined as TNM Stage I to IIIa, surgery is the main way for curative intent [66]. However, the 5 year survival rate drastically declines from the 92% for stage Ia to 36% for stage IIIa [67], indicating the presence micrometastasis at time of surgical resection. For patients with stage II and III adjuvant, cisplatin-based adjuvant chemotherapy (AC) improves survival, with a small 5-year absolute benefit of only 5.4% from chemotherapy; on the other hand, there are 0.9% deaths related to cisplatin-based adjuvant chemotherapy [68].

Deepening in the adjuvant setting, vinorelbine is the only third generation drug in combination with cisplatin, tested so far, to demonstrate a significant survival benefit with a reduction in the risk of death of 20% versus observation, despite an increase in toxicity. Overall reported grade ≥3 toxicity was observed in 90% of patients receiving cisplatin–vinorelbine versus 49% in cisplatin–other. A small increase in chemotherapy-related deaths in cisplatin–vinorelbine compared with cisplatin–other (1.4% versus 0.4%) [69]. According with these data, for every hundred patients treated with adjuvant chemotherapy there are 60% for whom the treatment would not be necessary, 36% for whom it is not effective and only 4% that has an effective benefit. Substantially, 96% of treated patients do not benefit from treatment, but are exposed to adverse events that sometimes are fatal [70]. There is an unmet need to better stratify early stage NSCLC patient before proposing AC to improve their outcome and reduce the amount of patients who do not really need further treatments other than surgery. Follow-up programs to detect disease recurrence early rely mainly on positron emission tomography (PET) and computed tomography (CT) [66]. Although the sensitivity and specificity of these conventional imaging studies has improved much over the years, there are still several limitations. In particular, the sensitivity and specificity of PET for the detection of both mediastinal and distant metastatic disease is 95% and 83%, respectively, with a sensible decrease for lesions that are less then 1 cm [71]. The effect of CT surveillance after surgical resection for stage I to IIA NSCLC steadily improved patients OS between 1998 and 2009 [72], even though both the aforementioned techniques are not capable of detecting MRD. On the other hand, serial postsurgical ctDNA analysis could identify recurrence/metastasis earlier than routine radiological imaging [59,73].

Tumor size and volume both in metastatic and early stage setting correlate with survival [74,75], suggesting that the least the tumor volume the higher the chance of treatment response. In the preoperative setting, the association of ctDNA detection and tumor stage is a strong predictor of relapse-free survival (RFS) and OS in localized NSCLC patients undergoing complete resection. Moreover, the postoperative detection of ctDNA is able to detect early relapse [76].

A dynamic study investigated the perioperative dynamic changes in ctDNA levels in surgical patients. The authors found a correlation between the tumor maximum square and the amount of ctDNA release in plasma, suggesting that the lower the tumor cell burden, the lower the capacity to detect the smallest stage I lung cancer that, as for guidelines, do not need ACT after surgery [66]. In this study, it has also been demonstrated that 3 days after surgery is a good time to collect plasma for MRD detection, and the RFS among patients who had positive ctDNA at 3 days after surgery was 270 days for those who underwent ACT and 111 days for those who did not [33].

Qiu et al. reported data from a prospective trial where ultradeep targeted next-generation sequencing was used to evaluate the clinical utility of ctDNA for dynamic recurrence risk and adjuvant chemotherapy benefit prediction in resected NSCLC. Starting from the diagnostic tissue biopsy to obtain a mutational profile, they collected plasma pre-surgery to identify potential clinicopathological determinants of ctDNA shedding. Almost 70% of patients had somatic mutations detectable in ctDNA, in rates of 61% for patients in stage I/II and 76% in stage III. Based on the histologic features, the ctDNA-positive rate was 46.1% for adenocarcinoma and 100% for all the other NSCLC subtypes including squamous cell carcinoma. Postsurgical ctDNA status suggests the possibility to stratify high-risk recurrence NSCLC patients in two groups: ctDNA-positive patients who could more likely benefit from ACT treatment and ctDNA-negative patients, where ACT seems to be unnecessary [70].

Similarly, the LUNGCA study evaluated the dynamic change of ctDNA in the perioperative setting for NSCLC patients, using a customized 769-gene panel. A total of 330 patients were evaluated in this study, and ctDNA in the preoperative setting was robust prognostic factor for RFS for adenocarcinomas whereas it was not significant for the squamous histology [77]. In the postoperative setting, ctDNA status was a strong survival biomarker for both adenocarcinoma and squamous carcinoma and among the classical clinicopathological variables (i.e., age, tumor size, histological subtype, TNM stage) ctDNA resulted in being the most reliable and independent risk factor for relapse-free survival. This study evaluated the role of cfDNA in detecting MRD in 261 patients with NSCLC in the perioperative setting, finding that AC conferred survival benefit to patients with detectable MRD after surgery, and that AC may be not necessary for undetectable MRD patients. Furthermore, the authors found that for the patients with brain-only relapse, the sensitivity of MRD monitoring is limited [78]. Similar results emerged from the GASTO1035 trial, where in five patients treated with AC the ctDNA status changed from positive to negative after treatment and only one patient experienced relapse [79].

The LUCID trial investigated the role of ctDNA in early-stage NSCLC patients treated with curative intent using a patient-specific assay. The authors found a positive prediction value of ctDNA detection for recurrence of 95% with a specificity of 98.7%. The ctDNA detection after treatment was associated with a 5.5-fold higher risk of death and 14.8-fold higher risk of recurrence [80].

The main clinical studies that included the study of ctDNA and survival of patients, as well as the possible benefit of ACT, are shown in Table 1.

## 4. Minimal Residual Disease in Locally Advanced NSCLC

Patients with locally advanced NSCLC unsuitable for a surgical approach can count on the multimodality approach of definitive chemoradiation; unfortunately, long-term survival is poor because sooner or later they experience disease relapse [67]. The PACIFIC trial demonstrated that immune checkpoint inhibition after a successful chemoradiotherapy can improve both progression-free survival (PFS) and OS, but unfortunately only for PD-L1-positive patients [81]. Moding et al. reported results of a retrospective trial in which they evaluated the role of ctDNA in this setting of patients [82]. All patients with undetectable ctDNA after chemoradiation treatment were free from progression at 24 months of follow-up. Interestingly, their results suggest that the administration of immune checkpoint inhibitors (ICI) after chemoradiation therapy could be avoided in negative ctDNA patients, as reported in the adjuvant setting. Crossing the data between the Pacific trial and Moding’s report, it emerges that we should treat 112 ctDNA-negative patients with ICI consolidation treatment after definitive chemoradiation to give benefit to a single patient, considering the 4% of Grade 5 events reported in the durvalumab arm of the pacific trial [82,83]. Moreover, several interventional studies (e.g., NCT04585477 and NCT04585490) consider ctDNA changes as a primary or secondary outcome measure, suggesting that ctDNA changes are a surrogate of clinical benefit [29]. The ongoing clinical trials investigating the ctDNA profiles for NSCLC patients are shown in Table 2.

## 5. Technical and Biological Factors

Several parameters have to be considered when analyzing ctDNA for lung cancer MRD detection. Timing of blood draw procession, as ctDNA half-life ranges between 16 min and 13 h, and timepoints for surveillance are crucial, as it has been demonstrated that post-surgery proinflammatory processes enhance cfDNA for up to one month, resulting in ctDNA levels below the LOD of the selected technique [84,85]. As mentioned, LOD is a main aim for detection of liquid biopsy techniques, and optimal pipelines for mutations calling in bioinformatics are needed to distinguish background signals of sequencing steps from true DNA mutations. Moreover, some technical errors could be introduced using NGS library preparation or sequencing steps in a range of 0.1–1% [86]. Most of these methods are based on molecular barcoding, where unique molecular sequences are ligated to targeted regions during library preparation to increase the potential sequenced reads, to reach a sequence consensus and increase sensitivity, while technical errors could be also reduced by computational biology improvements [28,54,55,87]. Molecular barcoding and computational biology improvements become essential, especially when augmenting sequencing depth, as library preparation errors could result in false positive higher rates.

In order to assess accuracy, sensitivity and reproducibility of ctDNA tests, the SEQC2 Oncopanel Sequencing Working Group compared five industry-leading ctDNA assays; they highlighted that ctDNA mutations are detectable with high sensitivity whether above 0.5% VAF [88]. A similar approach was adopted by Stetson and colleagues, who compared four plasma NGS tests, finding substantial variability in sensitivity (range 38–89%) and positive predictive values (36–80%), with highest levels of discordance observed in <1% VAF [89]. Taken together, these results highlight that both biological and technical factors affect error rates when profiling tumors using ctDNA analyses.

A main biological factor to take into consideration when profiling ctDNA is clonal hematopoiesis (CH) as a major source of false positives. CH is a biological process of acquisition of genomic somatic alterations in hematopoietic stem and/or progenitor cells, which are associated with smoking and radiation therapy and usually accumulate during aging [90]. These genomic alterations display clonal expansion and are associated with hematologic malignant neoplasms, therapy-related hematologic neoplasms, and are common in patients with solid cancers [91,92]. Data from 17,469 patients with 69 cancer types revealed that 14.1% of somatic alterations detected in hematopoietic cells is attributable to CH in patients with solid tumors, even at higher frequencies than tumor tissue and in cancer hotspots [90]. Another study showed that in patients with NSCLC, variants occurring in white blood cells (WBC) at a ≥2% VAF affect canonical CH genes, with respect to those at <2% VAF [60]. This biological background contributes to cfDNA false positive rates, and WBC sequencing could help to identify CH genomic alterations, which could be related to WBC but not to primary tumors, considering that 10% of alterations occur in the cfDNA of healthy subjects [60,93,94,95].

## 6. Conclusions

All these data support the utility of ctDNA testing to support clinical practice in decision-making for patient treatment. In the future, this sensitive tool could better identify patients at high risk of relapse who may benefit from adjuvant treatment, replacing and/or flanking the actual stage system. This would be a revolutionary step in the clinical management of resectable NSCLC patients in tailoring treatments and achieving the best therapeutic benefit. On the other hand, available data to date are quite heterogeneous in terms of used platform, timepoints for blood drawing and study endpoints, and the influence of technical and biological parameters could introduce a considerable issue to minimize. Moreover, still there is no consensus about the use of a tumor-informed or tumor non-informed approach.

The detection of MRD in NSCLC has to be reliable, with high sensitivity and acceptable turnaround times. For these reasons, the tumor-informed approaches brought the most interesting results, and this approach was demonstrated to be preferable, also for the relatively higher sensitivity. Moreover, in the earlier stages of malignancy, it is more likely to find commonly cancer-associated genomic alterations, and the use of a comprehensive or patient-designed panel could be more clinically useful, so a targeted sequencing approach would be preferable. At the current state of the art, patients are treated with AC only based on disease stage, and overtreatment for patients with negative MRD with ctDNA could be possible. On the other hand, for patients positive for ctDNA, who remain positive after AC, a stricter follow-up program could be imagined for ready administration of systemic therapy or local treatment of “oligo-progressions”.

To this aim, randomized trials based on ctDNA status are needed, as well as the most reliable ctDNA detection technologies.

Several prospective clinical trials are testing the clinical utility of ctDNA in addressing patients to adjuvant or consolidation therapy, and results could pave the way for the use of ctDNA in clinical settings.

## Figures and Tables

**Figure 1 life-13-01915-f001:**
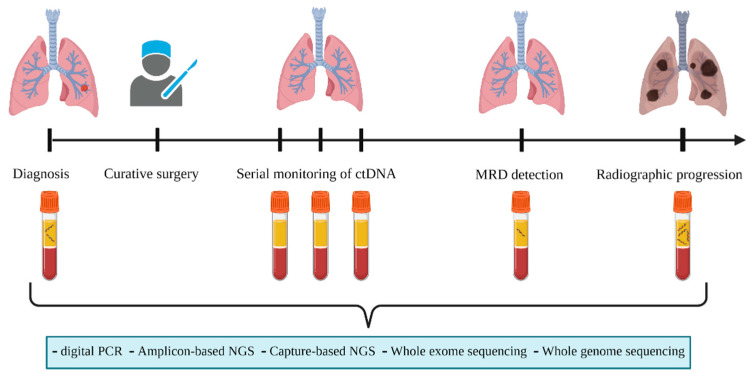
Schematic representation of serial monitoring of ctDNA during follow up in patients with early-stage non-small cell lung cancer for minimal residual disease detection before radiographic progression. The figure was created with BioRender (www.biorender.com, accessed on 1 June 2023).

**Table 1 life-13-01915-t001:** Studies analyzing ctDNA in relation to survival and adjuvant chemotherapy in resectable non-small cell lung cancer patients.

Study Name	Timepoints for ctDNA Analysis (Number of Patients)	ctDNA and Survival Prediction	Role of Adjuvant Chemotherapy	Methodology	Ref.
Dynamic	Perioperative (36)	Relation between TMs and ctDNA	Benefit from AC for ctDNA-positive patients	Targeted NGS technology (cSMART)	[33]
-	Postsurgery, post-ACT (103)	ctDNA positivity predicts worse RFS	Benefit of AC only in ctDNA-positive patients	Targeted custom 139 lung cancer-associated gene NGS panel	[70]
LUNGCA	Perioperative (330)	ctDNA positivity predicts worse survival	Benefit of AC only in ctDNA-positive patients	Targeted custom 769-gene NGS panel	[77]
LUCID	Preoperative and longitudinal (88)	95% PPV of ctDNA, with 98.7% specificity	-	Tumor-informed RaDaR patient-specific NGS panel using 48 variants	[80]
GASTO 1035, 1028	Preoperative and longitudinal (119)	ctDNA positivity predicts worse RFS	-	Targeted 425 cancer-related gene NGS panel	[79]

AC: adjuvant chemotherapy; PPV: positive predictive value; RFS: relapse-free survival; ctDNA: circulating tumor DNA; NGS: next-generation sequencing.

**Table 2 life-13-01915-t002:** Clinical trials investigating ctDNA profiles in resectable non-small cell lung cancer patients.

Identifier	Brief Summary	Study Type (Phase)	Primary Endpoint	Secondary Endpoints
NCT05286957	MRD-guided adjuvant tislelizumab + CT vs. standard adjuvant tislelizumab + CT	Interventional parallel assignment (II)	2-year PFS rate	Percentage of patients changed from MRD+ to MRD- after treatment with tislelizumab at 6–9–12 months
NCT03770299	Adjuvant nivolumab + SOC vs. SOC alone NSCLC patients with detected MRD	Interventional parallel assignment (II)	DFS at 24 months	ctDNA at response rate, duration of response and time to response; toxicity profile.
NCT04585477	Detect the number of circulating cancer cells after durvalumab after the standard treatment for patients positive for ctDNA	Interventional parallel assignment (II)	Changes in ctDNA from trial enrollment to after two cycles of adjuvant durvalumab	DFS, OS, toxicity profile
NCT04585490	Detect the number of circulating cancer cells after combination of the standard treatment (durvalumab) with additional chemotherapy	Interventional parallel assignment (III)	Changes in ctDNA levels after consolidation chemotherapy and immunotherapy	To determine the proportion of subjects in Cohort 1 MRD+ for whom ctDNA becomes undetectable after chemotherapy, OS, PFS, toxicity profile.
NCT04367311	Adjuvant chemotherapy + atezolizumab in stage I–III NSCLC who have detectable ctDNA after surgery	Interventional parallel assignment (II)	ctDNA levels after 4 cycles of adjuvant CT + atezolizumab plus up to 13 additional cycles of atezolizumab	ctDNA clearance in all therapeutic settings, one-year DFS

SOC: standard of care therapy; NSCLC: non-small cell lung cancer; MRD: minimal residual disease. ctDNA: circulating tumor DNA. DFS: disease-free survival; OS: overall survival; PFS: progression-free survival.

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
