# Peer review of "The Clinical Significance of Circulating Tumor DNA for Minimal Residual Disease Identification in Early-Stage Non-Small Cell Lung Cancer"

_life, 2023, doi:10.3390/life13091915_

Round 1

Reviewer 1 Report

The review article by Verlicchi et al titled " Clinical significance of ctDNA for MRD identification NSCLC" was not adding perspective on ctDNA in MRD detection. The review article was not adequately addressed any on the topic related to challenges and was not comprehensive on the ctDNA on NSCLC.

The current version is not acceptable for publication.

Author Response

The review article by Verlicchi et al titled "Clinical significance of ctDNA for MRD identification NSCLC" was not adding perspective on ctDNA in MRD detection. The review article was not adequately addressed any on the topic related to challenges and was not comprehensive on the ctDNA on NSCLC.

The current version is not acceptable for publication.

Re: We thank the Reviewer for the comments. We are sorry the Reviewer has found our manuscript as not comprehensive and not adequately addressed to the topic of ctDNA and NSCLC. As the Reviewer may note by the re-submitted version of the manuscript, we made extensive changes to our manuscript, even changing the order of paragraphs. All sections changed, also according to the comments and suggestion by other Reviewers; several aspects of the focus of the manuscript have been added and discussed, with relative bibliography, as well as the methodologies have been discussed from a technical point of view, also adding a Table.

We kindly invite the Reviewer to re-consider our manuscript in its reviewed version, with any comment or suggestion will be welcome to further improve our manuscript. 

Reviewer 2 Report

The clinical significance of circulating tumor DNA for Minimal Residual Disease identification in early-stage Non-Small Cell Lung Cancer 

The authors set out their aims for studying the applicability of ctDNA to determine MRD and prognosis in LC with a focus on clinical trials.

The authors start by providing some information about LC and relevant stats. This is followed by an introduction to NSCLC as the most prevalent type and states some detail about OS as per this disease. This is how I would write a review, with good detail about the disease and a logical flow.

The authors then move on to introducing cfDNA.

The authors could explain how they get the number 80 for a nucleosome. To my knowledge these are 200, 166 or 147 based on the level of digestion, 80 I have not heard before.

The 166 and 10kp size description is interesting. The location of cfDNA is then mentioned. The authors then narrow it down to ctDNA and state the issue; ctDNA clearance after therapy, this clearance issue could be elaborated on more in a few sentences.

Given the promising results of the liquid biopsy for the identification of ctDNA, several interventional trials are currently active that use the liquid biopsy to selector stratify patients (Table 1). This sentence needs more detail, for example, what do the authors mean by selector stratify? Table 1 could be explained more. In general, the authors could build up more momentum before adding this table.

2. The authors start with OS per stage of NSCLC and introduce micrometastasis at the time of surgery which may be overcome with treatment vinorelbine-cisplatin and related stats for that are introduced. The authors state the issue with this treatment and sometimes fatal adverse reactions in some patients.

don’t, I would advise against using contractions.

The authors mention early detection programmes using PET and CT. Relevant data for the sensitivity of these methods also follows and the authors mention the utility of ctDNA and how this might be a method of choice for detecting MRD beyond the imaging modalities.

The link between tumour size and volume with treatment and relapse is mentioned. Subsequently, the Dynamic, Qui, LUNGCA and LUCID studies have been covered to outline studies that focused on ctDNA, treatment and MRD. The authors could summarise this in a table. Ultimately detecting MRD after treatment was linked to a higher risk of death and recurrence. The studies in this section could be cross-compared like what you have done for section 3.

3. Links between MRD and local disease have been introduced through the PACIFIC trial. Please define ICI consolidation. The studies were cross-compared.

4. This could be a stylistic comment but shouldn’t section 4 proceed 2 and 3? Since it is introducing MRD detection? The authors outline the LOD definition and talk about a few detection methods. I’m not sure if this sentence is well placed here:

Moreover, several interventional studies (e.g. NCT04585477 and NCT04585490) consider ctDNA changes as a primary or secondary outcome measure, suggesting that ctDNA changes are a surrogate of clinical benefit [33].

4.1. crucial aspects of PCR have been mentioned, including the specificity of ddPCR.

While dPCR is very useful to… is this still ddPCR or has the technique changed? The authors mention the pros and cons of dPCR and associated panels for MRD detection. They could elaborate here on why setting up a panel is challenging. 

I’m also a little confused about this sentence: Consequently, this methodology is more suitable for oncogene addiction diagnosis, targeted therapy monitoring and early cancer progression identification, in this subset of patients. Why are these the key applications of this technique?

In this section, the authors could add another study using dPCR so they can compare. Also, kindly make sure you are not steering off-topic and mention a study that uses these techniques for some type of ctDNA detection application.

What has happened here [37039321]?

4.2. The lower sensitivity issue is mentioned at the start. How about a high-depth (high-pass) targeted sequencing? This is a good technique.

Generally, NGS methodology is considered to reach a lower sensitivity than dPCR, even though ultra-deep sequencing was revealed to reach 75% and 100% concordance with tissue sequencing, respectively, and to detect the same mutations results on EGFR and KRAS mutations with respect to dPCR in liquid biopsies in 21 out of 22 patients [49].  The structure of the sentence does not make it easy to link which technique is 75% and which is 100%.

Another approach tested the potential of molecular amplification pools (MAPs) in reducing NGS-sequencing errors, with a sensitivity of 98.5% and a specificity of 98.9%, with similar accuracy with respect to dPCR [31].  How do MAPs work, please explain.

In lines 199-216, the authors are getting a little sidetracked with the technical aspect but kindly add relevant information about MRD/ctDNA. Machine learning method/lung clip is also looking at ctDNA.

in particular, this method, namely PhasED-Seq, identified 25% of patients with 211 undetectable ctDNA by CAPP-seq, who had worse outcomes [57].  In general, it might be better to give fewer examples but thoroughly explain each example and not deal with it in passing. The authors swiftly move on to TEC-seq leaving the author confused about what all these sequencing methods are, how they are performed and how they differ (and what their pros and cons are).

Make sure all acronyms have been defined such as SNV.

Studies 10, 59 and 60 are relevant but could be cross-compared if possible.

In general, the NGS section is too complex and compact and could be broken down into more legible subsections, studies could be explained more and cross-compared when relevant. Techniques could be explained so the reader leaves with gaining some knowledge about the technique.

WGS ensues and pros and cons have been mentioned (please use this format for the multiple new sequencing techniques mentioned on page 5).

Most of ctDNA profiling approaches are based on the “tumor informed” analysis, which has the aim of detecting and monitoring genomic alterations firstly identified in tumor tissue. Even though the tracking of specific mutations would require less material and could take advantage of a more sensitive methodology, it could be less informative than “tumor non-informed” approaches [33]. I don’t really understand this paragraph, please explain this more.

I don’t understand what the blue box underneath figure 1 (the box outlining techniques) is referring to.

5. The technical limitations section is useful. On line 262, the oncopanel study is interesting.

A main biological factor to take into consideration when profiling ctDNA is clonal hematopoiesis (CH), as a major source of false positive. CH is a biological process of acquisition of genomic somatic alterations in hematopoietic stem and/or progenitor cells, which are associated to smoking and radiation therapy, and usually accumulate during aging [67]. How does this happen in non-haemotological cancers?

The authors could also mention methods to determine the percentage of ctDNA in whole cfDNA (tumour fraction) and how this may change in the timepoints mentioned in figure 1.

The authors also ultimately what is their recommendation for techniques to reliably detect ctDNA for MRD/ predicting relapse.

Some proof-reading required

Author Response

The clinical significance of circulating tumor DNA for Minimal Residual Disease identification in early-stage Non-Small Cell Lung Cancer 

The authors set out their aims for studying the applicability of ctDNA to determine MRD and prognosis in LC with a focus on clinical trials.

The authors start by providing some information about LC and relevant stats. This is followed by an introduction to NSCLC as the most prevalent type and states some detail about OS as per this disease. This is how I would write a review, with good detail about the disease and a logical flow.

Re: We really thank the Reviewer for appreciating the content of our general introduction as well as how we detailed NSCLC overview and stats.  

The authors then move on to introducing cfDNA.

The authors could explain how they get the number 80 for a nucleosome. To my knowledge these are 200, 166 or 147 based on the level of digestion, 80 I have not heard before.

Re: We thank the Reviewer for the comment. In the manuscript, we stated 80bp as a histone-associated length of cfDNA. We reported 80bp DNA fragment length as this was the lower length of histone-associated dsDNA (PMID: 30479833). As the Reviewer noticed, the common nucleosomes fragments are 200, 166 and 147 bp, and we accordingly modified our sentence for cfDNA length (PMID 26771485, 2669344). Moreover, we better described ctDNA dynamics such as the mechanisms of release of (PMID 26771485, 837366, 28577941) [lines 39-40].

The 166 and 10kp size description is interesting. The location of cfDNA is then mentioned. The authors then narrow it down to ctDNA and state the issue; ctDNA clearance after therapy, this clearance issue could be elaborated on more in a few sentences.

Re: We really thank the Reviewer for the comment. We also thought that it was interesting to include the different length patterns of cfDNA. We totally agree with the Reviewer that ctDNA clearance is a key point to be discussed, thus we deepened the matter conceirning the ctDNA clearance (PMID 27259816, 11245480, 18670422, 24553385, 30990132, 10444277) [lines 52-69].

Given the promising results of the liquid biopsy for the identification of ctDNA, several interventional trials are currently active that use the liquid biopsy to selector stratify patients (Table 1). This sentence needs more detail, for example, what do the authors mean by selector stratify? Table 1 could be explained more. In general, the authors could build up more momentum before adding this table.

Re: We totally agree and thank the Reviewer for the comment. With respect to Table 1, we agree that it has to be introduced after discussing the results of clinical trials assessing ctDNA. As a consequence, we deleted the entire sentence referring to Table 1 to put it later in the maintext, together with Table, which has been moved later in the text. As a detail for the Reviewer, as a “selector stratify”, it was meant that some studies are randomising patients basing on the results of ctDNA to assess MRD after curative treatment, and it was defined the liquid biopsy as the clinical tool to stratify patients for treatment selection. Together with these changes, we better introduced the role of ctDNA in advanced clinical setting and for MRD detection in early stage and locally advanced NSCLC (added PMID 37173891, 36357680, 32206559, 36928816, 34985936, Lebow ES J. Clin. Oncol. 2022, 40, 8540, doi:10.1200/JCO.2022.40.16_suppl.8540) [lines 66-71].

  1. The authors start with OS per stage of NSCLC and introduce micrometastasis at the time of surgery which may be overcome with treatment vinorelbine-cisplatin and related stats for that are introduced. The authors state the issue with this treatment and sometimes fatal adverse reactions in some patients.

don’t, I would advise against using contractions.

Re: We thank the Reviewer for noticing the informal use of the verb, we accordingly corrected it [line 312]

The authors mention early detection programmes using PET and CT. Relevant data for the sensitivity of these methods also follows and the authors mention the utility of ctDNA and how this might be a method of choice for detecting MRD beyond the imaging modalities.The link between tumour size and volume with treatment and relapse is mentioned. Subsequently, the Dynamic, Qui, LUNGCA and LUCID studies have been covered to outline studies that focused on ctDNA, treatment and MRD. The authors could summarise this in a table. Ultimately detecting MRD after treatment was linked to a higher risk of death and recurrence. The studies in this section could be cross-compared like what you have done for section 3.

Re: We thank the Reviewer for the comment. We agree that these studies should be cross-compared and that a Table resuming the results would be more informative. We accordingly added a Table including the mentioned studies and the GASTO 1035 and 1028, to directly display and cross-compare the studies [Table 1].

  1. Links between MRD and local disease have been introduced through the PACIFIC trial. Please define ICI consolidation. The studies were cross-compared.

Re: We thank the Reviewer for the comment. ICI consolidation is meant for the use of immune checkpoint inhibitors after chemo-radiation therapy, we accordingly modified [line 389].

  1. This could be a stylistic comment but shouldn’t section 4 proceed 2 and 3? Since it is introducing MRD detection? The authors outline the LOD definition and talk about a few detection methods. I’m not sure if this sentence is well placed here:

Moreover, several interventional studies (e.g. NCT04585477 and NCT04585490) consider ctDNA changes as a primary or secondary outcome measure, suggesting that ctDNA changes are a surrogate of clinical benefit [33].

Re: We thank the Reviewer for the comment. Taking a deeper look to our manuscript, we agree that the section 4 should proceed sections 2 and 3. In fact, this section introduced MRD and techniques to detect it, and it would be useful for the reader to receive a clear description of the object of the Review and how MRD is achieved by liquid biopsy techniques. It is now numbered as section 2. We also agree with the Reviewer that the mentioned sentence is not well placed in the paragraph, and it was moved to the clinical section [lines 394-398].

4.1. crucial aspects of PCR have been mentioned, including the specificity of ddPCR.

While dPCR is very useful to… is this still ddPCR or has the technique changed? The authors mention the pros and cons of dPCR and associated panels for MRD detection. They could elaborate here on why setting up a panel is challenging. 

Re: We thank the Reviewer for the comment. The technique was not changed and it still was ddPCR. This was fixed [lines 100, 102]. We agree with the Reviewer than in this part the reasons why setting up a dPCR panel is difficult. While only some aspects has been already highlighted (i.e. “realtively high costs and it needs highly-specific probes”), we have now better focused on the difficulties of the methodology in setting up panels for MRD through liquid biopsy (added PMID 33257044, 28233803), especially in terms of knowing which mutations have to be detected and about the multiplexing different probes [lines 133-138].

I’m also a little confused about this sentence: Consequently, this methodology is more suitable for oncogene addiction diagnosis, targeted therapy monitoring and early cancer progression identification, in this subset of patients. Why are these the key applications of this technique?

Re: We thank the Reviewer and appreciate the comment, as we have the possibility to better clarify the sentence. With this comment, we wanted to underline that the dPCR better express its potential, mainly based on the high sensitivity of the technique, when applied to liquid biopsy for the detection of activating mutations (eg EGFR, “oncogene addiction diagnosis”), resistance mutations to tyrosine kinase inhibitors (eg ALK C1156Y, I1171N or EGFR C797X, “targeted therapy monitoring”) and “early cancer progression identification”, given the possibility to longitudinally monitor plasma samples of an oncogene addicted NSCLC patient to early identify resistance mutations to targeted therapy and cancer progression. In fact, these mutations are usually clonally expressed within the tumor mass, and liquid biopsy is a non invasive tool to early identify known hotspot mutations, while the dPCR is one of the most sensitive methodologies able to detect low allele frequencies. In fact, as explained within the manuscript, dPCR is more suitable to search known mutations (as activating or resistance mutations to tyrosine kinase inhibitors), and multiplexing is often challenging; these impressions have been also highlighted by other authors, as Gassa and colleagues stated that “Knowing the gene mutation is mandatory to perform ddPCR in plasma“ (PMID 33257044), while Wan and colleagues stated that “These are, therefore, generally suited to investigating a small number of mutations and are often applied to analysis of cancer hot-spot mutations” (PMID 28233803). At this regards, Peng and colleagues pointed that “Although ARMS, ddPCR, and BEAMing have excellent sensitivity and detection capabilities for various stages of cancer, their clinical applicability is restricted since these methods can only identify known mutations” (PMID 34868984). We thank the Reviewer and we have accordingly modified the sentence to make it clearer [lines 138-141].

In this section, the authors could add another study using dPCR so they can compare. Also, kindly make sure you are not steering off-topic and mention a study that uses these techniques for some type of ctDNA detection application.

Re: We thank the Reviewer for the comment. We agree that another study using this technique for ctDNA detection application would be more informative for the reader. Accordingly, we added a study by Xu and colleagues detecting MRD by a PCR-based approach (PMID 35308511) [lines 181-185]. At the best of our knowledge, still there are no studies using only ddPCR to detect MRD, as most of studies performed an NGS approach before using ddPCR for mutations in ctDNA At this regard, we discussed these studies in the 2.2 section [lines 152-263].

What has happened here [37039321]?

Re: Many thanks to the Reviewer for the noticing the error. This was a PMID of a paper by Zhao Y and colleagues that we wanted to cite, but it was not inserted as a reference. It has now been fixed [line 148].

4.2. The lower sensitivity issue is mentioned at the start. How about a high-depth (high-pass) targeted sequencing? This is a good technique.

Re: We thank the Reviewer for the comment. In this section, all targeted sequencing panels we mention are high-depth panels; in fact, considering the needed limit of detection for MRD assessment by liquid biopsy, all targeted panels need to have a high coverage. We agree with the Reviewer that these are very good techniques, and we notice that augmenting the sequencing depth is not synonym of obtaining a better sensitivity. To this matter, we highlighted in Section 5 (Technical and biological factors) that molecular barcoding, such as computational biology improvements, are essential to eliminate PCR duplicates and NGS library preparation errors, especially when augmenting sequencing depth. In this section, we better highlighted this matter [lines 424-426].

Generally, NGS methodology is considered to reach a lower sensitivity than dPCR, even though ultra-deep sequencing was revealed to reach 75% and 100% concordance with tissue sequencing, respectively, and to detect the same mutations results on EGFR and KRAS mutations with respect to dPCR in liquid biopsies in 21 out of 22 patients [49].  The structure of the sentence does not make it easy to link which technique is 75% and which is 100%.

Re: We really thank the Reviewer for noticing the error in the sentence. We accordingly corrected the sentence [lines 168-169].

Another approach tested the potential of molecular amplification pools (MAPs) in reducing NGS-sequencing errors, with a sensitivity of 98.5% and a specificity of 98.9%, with similar accuracy with respect to dPCR [31].  How do MAPs work, please explain.

Re: We thank the Reviewer for the comment. We agree that a custom methodology has to be better detailed. We added a brief technical description of the main advantages of the methodology [lines 177-180].

In lines 199-216, the authors are getting a little sidetracked with the technical aspect but kindly add relevant information about MRD/ctDNA. Machine learning method/lung clip is also looking at ctDNA.

Re: We thank the Reviewer for the comment. We agree that technical aspects could be sidetracking within a manuscript which is mainly focused to the clinical potential of a laboratory approach; on the other hand, for MRD identification by NGS, technical aspects are a challenge for molecular biologists and bioinformaticians to reach reliable data to be useful in clinical trials or clinical practice.

in particular, this method, namely PhasED-Seq, identified 25% of patients with 211 undetectable ctDNA by CAPP-seq, who had worse outcomes [57].  In general, it might be better to give fewer examples but thoroughly explain each example and not deal with it in passing. The authors swiftly move on to TEC-seq leaving the author confused about what all these sequencing methods are, how they are performed and how they differ (and what their pros and cons are).

Re: We really thank the Reviewer for the constructing comment. We totally agree that a rapid passing through different studies could confound the reader. Our main focus was to present the results of each study, giving an overview of which are the potentials of different approaches and methodologies. We accordingly modified the part of the section focused on PhasED-Seq and TEC-Seq, describing technical aspects of the different approaches and discussing the pros and cons of the methodologies [lines 208-215; 219-224].

Make sure all acronyms have been defined such as SNV.

Re: We thank the Reviewer for noticing this error, we accordingly modified and defined the acronym SNV [line 226].

Studies 10, 59 and 60 are relevant but could be cross-compared if possible.

Re: We thank the Reviewer for the comment. These studies are not directly comparable, as two of them (Abbosh et al and Waldeck et al) are original articles, while the study by Verzè et al is a systematic review that includes the two studies in the analysis. Anyway, we thank the Reviewer because we changed the sentence discussing data for the study by Weldeck for the important data about PFS and OS [lines 235-236].

In general, the NGS section is too complex and compact and could be broken down into more legible subsections, studies could be explained more and cross-compared when relevant. Techniques could be explained so the reader leaves with gaining some knowledge about the technique.

Re: We thank the Reviewer for the comment. We agree that the NGS section was too complex, and we accordingly divided in two main sections (i.e. 2.2.1 Targeted sequencing, and 2.2.2 Whole Genome Sequencing and Whole Exome Sequencing), also adding an introduction part for NGS methodologies. We agree that a more comprehensive explication of the techniques could be more informative for the reader, and we accordingly introduced technical details for the the two most innovative and custom techniques, i.e. PhasED-Seq and TEC-Seq [lines 208-215; 219-224], as the Reviewer noticed; the other techniques are really standardized and common ones, and a brief introduction has been done (e.g. lines 198-199 for amplicon-based NGS).

WGS ensues and pros and cons have been mentioned (please use this format for the multiple new sequencing techniques mentioned on page 5).

Re: We thank the Reviewer for the comment. We agree that pros and cons also for targeted sequencing methodologies could be more informative for the reader. We accordingly used the same format for the targeted sequencing techniques [lines 239-244].

Most of ctDNA profiling approaches are based on the “tumor informed” analysis, which has the aim of detecting and monitoring genomic alterations firstly identified in tumor tissue. Even though the tracking of specific mutations would require less material and could take advantage of a more sensitive methodology, it could be less informative than “tumor non-informed” approaches [33]. I don’t really understand this paragraph, please explain this more.

Re: We thank the Reviewer for the comment. We agree that the sentence is not clear, so we better explained which is the significance of “tuomr informed” and “tumor non-informed” approaches, highlighting pros and cons and possible applications of each of the two approaches. We highlight that after the division of the paragraph in subsections, this part has been moved to the introduction of the paragraph [lines 157-166].

I don’t understand what the blue box underneath figure 1 (the box outlining techniques) is referring to.

Re: We thank the Reviewer for the comment. The blue box indicates all the methodologies that could be applied to a longitudinal analysis of ctDNA samples (see brackets) and there are discussed along the manuscript.

  1. The technical limitations section is useful. On line 262, the oncopanel study is interesting.

A main biological factor to take into consideration when profiling ctDNA is clonal hematopoiesis (CH), as a major source of false positive. CH is a biological process of acquisition of genomic somatic alterations in hematopoietic stem and/or progenitor cells, which are associated to smoking and radiation therapy, and usually accumulate during aging [67]. How does this happen in non-haemotological cancers?

Re: We thank the Reviewer for the comment. As highlighted by Ptashkin and colleagues (29872864), clonal hematopoiesis affects patients with different types of cancer, and the 14% of somatic alterations are attributable to such process. This also happens in non-haematological malignancies, as it is related to aging and smoking habits, and to radiation therapy (often used in patients with solid tumors).

The authors could also mention methods to determine the percentage of ctDNA in whole cfDNA (tumour fraction) and how this may change in the timepoints mentioned in figure 1.

Re: We thank the Reviewer for the comment. We have highlighted that tumor fraction is identified by mutant allele fraction found in the cfDNA, and that some authors also used a relation between MAF and ctDNA to calculate tumor fraction in the cfDNA. The ctDNA is depicted in Figure 1 as different quantities of DNA in the plasma from patients [lines 86-89].

The authors also ultimately what is their recommendation for techniques to reliably detect ctDNA for MRD/ predicting relapse.

Re: We thank the Reviewer for the comment. We agree that the opinion of the authors is essential in a Review article, and we accordingly added the authors’ suggestions for detection MRD [559-573].

We would really thank the Reviewer for the constructive, copious and valuable comments and suggestions to our manuscript. We truly believe that the Reviewer performed a really accurate revision that largerly helped to improve our Review.

Reviewer 3 Report

Review comments for “The clinical significance of circulating tumor DNA for Minimal 2 Residual Disease identification in early stage Non-Small Cell 3 Lung Cancer”

Summary:

This is a great review. The authors not only introduced the role of ctDNA in identifying MRD and in predicting prognosis of lung cancer patients, but also showed the limitation of ctDNA as a clinical practice biomarker. In the meantime, the authors provided the future directions for the future research in the field which may contribute to exploring the potential adjuvant or consolidation therapy in clinical.

Major Comment:

1.       The review showed that ctDNA testing is a highly sensitive and accurate test, which can improve the 5-year survival rate of lung cancer patients when used as an adjuvant or consolidation therapy. The purpose of this test is to detect minimal residual disease, meaning small tumor or cancer cells that couldn't be detected by other tests. However, clinical therapy for lung cancer cannot eliminate all lung cancer cells, implying that there is still a possibility that ctDNA testing can detect these remaining cancer cells. Does this mean we need to continue the treatment? Are you concerned that over-treatment may cause more pain for patients?

2.       There is no description about the cost per ctDNA testing. It is important factor for clinical therapy. It would be beneficial to add it.

Minor comments:

1.       Line 117 and 118: There is a large space between line117 and 118. It would be beneficial to delete it.

Author Response

Review comments for “The clinical significance of circulating tumor DNA for Minimal 2 Residual Disease identification in early stage Non-Small Cell 3 Lung Cancer”

Summary:

This is a great review. The authors not only introduced the role of ctDNA in identifying MRD and in predicting prognosis of lung cancer patients, but also showed the limitation of ctDNA as a clinical practice biomarker. In the meantime, the authors provided the future directions for the future research in the field which may contribute to exploring the potential adjuvant or consolidation therapy in clinical.

Major Comment:

  1. The review showed that ctDNA testing is a highly sensitive and accurate test, which can improve the 5-year survival rate of lung cancer patients when used as an adjuvant or consolidation therapy. The purpose of this test is to detect minimal residual disease, meaning small tumor or cancer cells that couldn't be detected by other tests. However, clinical therapy for lung cancer cannot eliminate all lung cancer cells, implying that there is still a possibility that ctDNA testing can detect these remaining cancer cells. Does this mean we need to continue the treatment? Are you concerned that over-treatment may cause more pain for patients?

Re: We thank the Reviewer for the comment. This is exactly what we meant in lines 284-288. At the current state of art, adjuvant chemotherapy is administered exclusively depending on the stage of disease, and after 4 cycles of therapy, the patient enters a follow up program. What we suggest is that patients over-treating could be avoided by the use of ctDNA, when ctDNA is unable to detect MRD, non-depending on the disease stage. On the other hand, for patients who are positive for MRD by ctDNA, and remain positive after adjuvant treatment, it may be possible to set a different follow up program, maybe more intensive, to be able to start a first-line treatment or a local treatment for “oligo-progressions”. Based on literature data, the less is the amount of disease, the higher possibility to achieve complete response. We better highlighted this point in the conclusion paragraph [lines 567-573].

  1. There is no description about the cost per ctDNA testing. It is important factor for clinical therapy. It would be beneficial to add it.

Re: We thank The Reviewer for the comment. We agree with the Reviewer that the precise cost per test would be beneficial for the reader. On the other hand, it would be really difficult to define an actual cost, as it depends on the number of tests performed (even with the same kit), the instrumentation used (eg using the same kit on different NGS platforms, eg MySeq, NextSeq or NovaSeq), the punctual discounts, the different costs across different countries. As a consequence, the most informative matter about the costs is to compare the different methodologies, discussing which methodologies could be more convenient, and to indicate a possible high cost as a limitation for a given methodology. Along the manuscript, this is what we tried to discuss for the readers.

Minor comments:

  1. Line 117 and 118: There is a large space between line117 and 118. It would be beneficial to delete it.

Re: We really than the Reviewer for noticing it. The large space between line 117 and 118 was deleted.

Round 2

Reviewer 2 Report

The authors have carefully addressed my comments.